# River network and hydro-geomorphological parameters at 1/12° resolution for global hydrological and climate studies

Simon Munier[1] and Bertrand Decharme[1]

[1]CNRM, Université de Toulouse, Météo-France, CNRS, Toulouse, France

**Correspondence:** S. Munier (simon.munier@meteo.fr)

**Abstract.** Global scale river routing models (RRMs) are commonly used in a variety of studies, including studies on the impact of climate change on extreme flows (floods and droughts), water resources monitoring or large scale flood forecasting. Over the last two decades, the increasing number of observational datasets, mainly from satellite missions, and the increasing computing capacities, have allowed better performances of RRMs, namely by increasing their spatial resolution. The spatial

resolution of a RRM corresponds to the spatial resolution of its river network, which provides flow direction of all grid cells. River networks may be derived at various spatial resolution by upscaling high resolution hydrography data. This paper presents a new global scale river network at 1/12° derived from the MERIT-Hydro dataset. The river network is generated automatically using an adaptation of the Hierarchical Dominant River Tracing (DRT) algorithm, and its quality is assessed over the 70 largest basins of the world. Although this new river network may be used for a variety of hydrology-related studies, it is here

provided with a set of hydro-geomorphological parameters at the same spatial resolution. These parameters are derived during the generation of the river network and are based on the same high resolution dataset, so that the consistency between the river network and the parameters is ensured. The set of parameters includes a description of river stretches (length, slope, width, roughness, bankfull depth), floodplains (roughness, sub-grid topography) and aquifers (transmissivity, porosity, sub-grid topography). The new river network and parameters are assessed by comparing the performances of two global scale

simulations with the CTRIP model, one with the current spatial resolution (1/2°) and the other with the new spatial resolution (1/12°). It is shown that CTRIP at 1/12° overall outperforms CTRIP at 1/2°, demonstrating the added value of the spatial resolution increase. The new river network and the consistent hydro-geomorphology parameters, freely available for download from Zenodo (https://doi.org/10.5281/zenodo.6482906), may be useful for the scientific community, especially for hydrology and hydro-geology modelling, water resources monitoring or climate studies.

## 1 Introduction

Global scale river routing models (RRMs) were primarily developed for climate studies. By simulating the flow routing processes through river networks, they allow climate models to close the water budget at the global scale. Then, several applications have been developed based on RRMs, including studies on the impact of climate change on extreme flows (floods and droughts, see e.g., Hirabayashi et al. (2013); Yamazaki et al. (2018)), water resources monitoring (e.g., Makungu and Hughes , 2021)

or large scale flood forecasting (e.g., GloFAS, Alfieri et al. , 2013; Jafarzadegan et al. , 2021).

Over the last two decades, the increasing number of observational datasets, mainly from satellite missions, and the increasing computing capacities, have allowed better performances of RRMs, either by improving the representation of some processes (e.g., Arora and Boer , 1999; Decharme et al. , 2008; Getirana et al. , 2021; Guimberteau et al. , 2012; Schrapffer et al. , 2020; Vergnes et al. , 2014; Yamazaki et al. , 2013), by integrating new ones, such as lake dynamics (Guinaldo et al. , 2021; Tokuda et al. , 2021) or dams operations (Dang et al. , 2020; Zajac et al. , 2017), or by increasing the spatial resolution. Several studies have demonstrated the benefit of increasing the spatial resolution in macroscale RRMs. For example, Mateo et al. (2017) showed that the river connectivity is better described at high spatial resolution, which improves the representation of the river flow dynamics within the river network. Nguyen-Quang et al. (2018) concluded that high streamflow simulation performances require a precise river catchment description, along with accurate forcing data (namely precipitation).

The river network, that mainly provides the flow direction of each cell, is the main component of a RRM. Higher spatial resolution allows to represent narrower rivers and to better localize confluences, with potential positive impacts on streamflow simulations. The river networks of most RRMs are either grid-based or vector-based. Both approaches differ in their definition of unit-catchments. In grid-based approaches, the river network is discretized on a regular Cartesian grid, so that unit-catchments are rectangular pixels. On the other hand, vector-based river networks are based on irregular shapes of unit catchments extracted from high resolution hydrography data. For instance, TRIP (Oki and Sud , 1998), CTRIP (Decharme et al. , 2019) and LISFLOOD (Van Der Knijff et al. , 2010) follow a grid-based approach, while CaMa-Flood (Yamazaki et al. , 2013), MGB-IPH (Collischonn et al. , 2007), VIC (We et al. , 2014) and RAPID (Lin et al. , 2018) follow a vector-based approach.

For grid-based models, the spatial resolution is defined by the size of the grid pixels, while for vector-based models, the spatial resolution relies on a threshold catchment area. For both approaches, the river network is generally derived from the upscaling of high resolution hydrography data. The HydroSHEDS dataset (Lehner et al. , 2008) has been the basis for a lot of upscaled river networks used in RRMs. Recently, Yamazaki et al. (2019) released a new hydrography dataset, MERIT-Hydro, based on the Multi-Error-Removed Improved-Terrain DEM (MERIT DEM, Yamazaki et al. , 2017) dataset. MERIT-Hydro has been used in a large number of recent studies (see, e.g., Lin et al. , 2019; Shin et al. , 2020; Eilander et al. , 2021; Getirana et al. , 2021), demonstrating its overall high quality for use in RRMs, among other purposes.

Generally, grid-based approaches follow a D8 convention, meaning that each grid cell may flow into one of the eight neighbouring grid cells. Vector-based approaches are more flexible and may follow a D∞ convention, for which the water in a unit catchment may flow into any other unit-catchment (not necessarily a neighbouring one). This is particularly convenient when two rivers flow through the same grid cell without being connected. The vector-based approach allows a better representation of sub-basins than the grid-based approach does, leading to increasing modelling performances (Yamazaki et al. , 2013). Yet, it is expected that the difference between both approaches should decrease when the spatial resolution increases. Moreover, grid-based RRMs are more easily coupled to Land Surface Models which generally also follow a grid-based approach. Under these considerations, it seems worthy to continue developing high spatial resolution grid-based river networks.

Besides, along with the river network at the appropriate spatial resolution, RRMs also require parameters that are consistent with the river network. Some parameters depend on the river network itself, such as length and slope of river stretches, and

vary with the spatial resolution. Other parameters, including roughness coefficient, river width or bankfull depth, may be calibrated or estimated using empirical relationships. In the latter case, these parameters may also indirectly depend on the spatial resolution. Finally, several RRMs use sub-grid approximations to represent fine scale processes. For instance, some flooding scheme (as, e.g., in Decharme et al. , 2008; Yamazaki et al. , 2011; Decharme et al. , 2012) relies on Cumulative Distribution Functions (CDFs) of flood volume and depth within each grid cell. Vergnes and Decharme (2012) also used CDFs for a sub-grid representation of groundwater dynamics. Such CDFs are computed from high resolution topography data and also directly depends on the spatial resolution of the RRM.

Although some recent studies provide new upscaled river network based on MERIT-Hydro (see, e.g., Eilander et al. , 2021), only a limited set of hydro-geomorphology parameters consistent with the new river network have been derived (such as sub-grid river length and slope). In this study, we propose to apply the Hierarchical Dominant River Tracing (DRT, Wu et al. , 2011) algorithm on MERIT-Hydro to derive a new global scale river network at 1/12° (5 arcmin) along with a set of consistent hydro-geomorphological parameters. The choice of 1/12° as the spatial resolution for river routing modelling is a compromise between a fine scale representation of river dynamics and computing efficiency. It is also well suited for global to regional scale studies. New features presented in this paper then include:

- river network at 1/12°
- river geomorphology (length, slope, depth, roughness)
- floodplains roughness and sub-grid topography
- aquifers characteristics and sub-grid topography

A direct quantitative assessment is not possible since there is, to our knowledge, no equivalent existing dataset at the same spatial resolution. As a consequence, to evaluate the new river network and the corresponding hydro-geomorphological parameters, we propose to use the CTRIP model (Decharme et al. , 2019) and to compare results of two global scale simulations: the first one at the CTRIP current spatial resolution (1/2°) and the second one at the new spatial resolution (1/12°). The CTRIP model has been chosen because of its efficiency, robustness and overall performances (see, e.g., Schellekens et al. , 2017; Decharme et al. , 2019). In addition, it was used in many global hydrological applications, some of which highlighting important results regarding global land hydrology (Douville et al. (2013); Cazenave et al. (2014); Padrón et al. (2020)). Whatever, the river network and parameters provided in this study could benefit to other similar large scale river routing models, or to any other study requiring all or part of this dataset at a similar fine spatial resolution.

The main purpose of this paper is to present the global river network at 1/12° and corresponding consistent hydro-geomorphological parameters. This dataset is mainly designed for all global or regional scale grid-based RRMs, although it could be used in a variety of hydrology-related studies that need flow direction at a medium spatial resolution (e.g., Catalán et al. , 2016; Robinne et al. , 2018; Scherer et al. , 2018; Wan et al. , 2015; Zhou et al. , 2015). A majority of large-scale RRMs uses a gridded structure for global hydrological studies (see technical review of Kauffeldt et al. , 2016) and most of them are still running at a coarse spatial resolution. So with the entire dataset described here (flow direction, river length, river slope, river bank-full depth, river roughness, floodplains roughness, major groundwater basins boundaries, aquifer transmissivity, and aquifer effective porosity),

many hydrological models could improve their river routing module by increasing the spatial resolution. Moreover, this consistent and comprehensive dataset can help modellers to integrate some important processes (such as inundation and groundwater) that are still neglected in some models.

The paper is organized as follows. The derivation of the river network at 1/12° is described in section 2, which also provides a quality assessment. Section 3 describes how hydro-geomorphological parameters are derived, while section 4 presents the results of CTRIP simulations at 1/2° and 1/12°, with comparisons with a large dataset of gauged river discharges.

## 2 River network at 1/12° resolution

This section describes the methodology to derive the river network at 1/12° resolution at the global scale.

### 2.1 Background

River network datasets consist in flow direction maps generally derived from Digital Elevation Models (DEMs) corrected for hydrology. With the increasing amount of satellite observations during the last decades, several methods have been proposed to derive river networks at various spatial resolutions using upscaling algorithms (for the D8 method, see, e.g., Döll and Lehner , 2002; Reed , 2003; Shaw et al. , 2005; Paz et al. , 2006; Davies and Bell , 2009; Wu et al. , 2011). All of them are based on a high resolution DEM and apply different upscaling strategies. Among them, the Hierarchical Dominant River Tracing (DRT, Wu et al. , 2011) presents interesting features for deriving D8 river networks. First it has been applied at different final resolutions (from 1/16° to 2°), showing its flexibility. Then it is a fully automated algorithm and does not necessitate any manual correction. Finally, it is designed to preserve the river network structure by processing major rivers first and applying river diversion when necessary. DRT has been applied by Wu et al. (2011, 2012) using the high resolution hydrography network from HydroSHEDS (Lehner et al. , 2008) and HYDRO1K (U.S. Geological Survey , 2000).

Recently, the Multi-Error-Removed Improved Terrain DEM (MERIT-DEM) has been proposed by Yamazaki et al. (2017). MERIT-DEM relies on the SRTM3 DEM (Farr et al. , 2007) and the AWE3D-30 m DEM (Tadono et al. , 2015) and integrated a variety of filters and ancillary datasets to remove major height error components. MERIT-DEM has been used to derive a high resolution (3 arc sec, about 90 m at the equator) global flow direction map, MERIT-Hydro (Yamazaki et al. , 2019), with good agreement with existing hydrography datasets such as HydroSHEDS (in terms of flow accumulation, river basin shape and river streamlines localization) and even significant improvements in some regions. Although MERIT-DEM and MERIT-Hydro are quite recent, they have been used in a large number of recent studies (see, e.g. Lin et al. , 2019; Moudrý al. , 2018; Shin et al. , 2019, 2020; Wing et al. , 2020), generally showing the added value of these datasets.

Here we applied the DRT algorithm using MERIT-Hydro as the basis for the high resolution hydrography network to make benefit of the most recent available dataset.

The following notations and definitions are used throughout the article:

- HR for high resolution (1/1200°) of MERIT
- 12D for 1/12° resolution

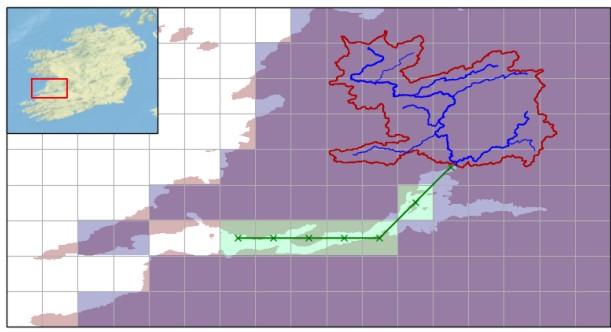

**Figure 1.** Example of estuary opening: red mask is the HR land mask, blue mask the 12D land mask, green mask represent the 12D cells converted from land to ocean to connect the river basin delineated in red to the ocean.

- HD for half-degree resolution
- pixel: unit element at HR
- cell: unit element at 12D

## 2.2 Methodology

The first step in the generation of the river network is the set up of a land mask at the final resolution (1/12°, thereafter denoted 12D). The land mask is used to ensure that no flow direction is given to cells in the ocean, which can happen during the diversion step (see bellow). The 12D mask relies on the HR mask from MERIT-Hydro. Cells are considered as land if at least 50 % of HR pixels within the cell are land pixels.

Particular attention has been paid to estuaries and their effective connection to oceans and seas. For example, it may happen that a large river flow into a narrow estuary. In this case, the cell corresponding to the river outlet may be disconnected to the coast by cells considered as land (more than 50 % of HR land pixels). To ensure an effective connection to the coast, closed seas (water cells surrounded by land cells) counting less than 20 cells are first converted to land. Then the HR land mask is used to find the shortest way within the estuary from the river outlet to cells marked as ocean, then cells covering this way are forced to ocean. In this process, only rivers with a flow accumulation greater than 10,000 pixels (HR) are considered. An example of an estuary in Ireland is presented in Figure 1.

Using the land mask as a basis, the DRT algorithm is applied to upscale the MERIT-Hydro river network from 3 arcsec to 1/12° resolution. Details of the DRT algorithm may be found in Wu et al. (2011, 2012). The main steps are reminded thereafter. Fig. 2 illustrates the process for the first largest rivers of the Hérault basin (France).

1. Rivers are first sorted by decreasing flow accumulation at the outlet. Rivers are treated in this hierarchical order to ensure the best representation of large rivers as possible. The following steps are applied for each river.

2. Given the river outlet, the HR river route is defined as the longest upstream river. The headwater cell is given by the first cell with a flow accumulation larger than a given threshold (10 % of cell size, i.e. 1000 HR pixels).

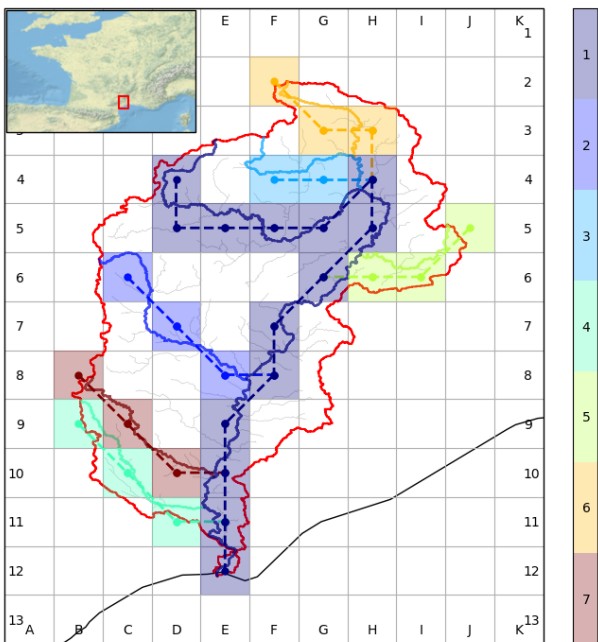

**Figure 2.** Example of network upscaling in the Hérault basin (France). Basin boundaries are drawn in red. Rivers are treated in descending order of their drainage area and drawn with different colors with solid lines for HR and with dashed lines for 12D.

3. Flow direction of the river at 1/12° is defined from the upstream cell to the outlet.

4. For each cell, the downstream cell is found when HR route exits the cell with a minimum length (60 % of cell size if cardinal, 80 % if diagonal, see, e.g., cell C7 for river #2 in Fig. 2). Each time a flow direction is assigned, potential intersections are checked and corrected if necessary.

5. If a downstream cell is already assigned (e.g., by a larger river), the river is diverted: a parallel route is created, as closed to the real river as possible, until HR route reaches an unassigned cell or outlet cell.

6. If the outlet is reached, the presence of loops is checked and corrected if necessary and the next largest river is treated (steps 2-6).

River diversion (step 5) is an important feature of the algorithm as it allows to conserve the structure of the network. But it simultaneously may raise problems with changes of river location (eg localization of gage station). To overcome this issue, it may be useful to keep a track of the relationship between HR and 12D rivers, which is done here by identifying each processed river with a unique id in both the HR and 12D networks. Note that while river diversion is necessary with a D8 convention, it can be avoided with a D∞ convention. Also, river diversion may have an impact on the attribution of input fluxes such as runoff used to force the routing model. Yet, we estimate that this impact could be neglected as runoff is generally a quite smooth field at this resolution when modelled by a Land Surface Model (LSM).

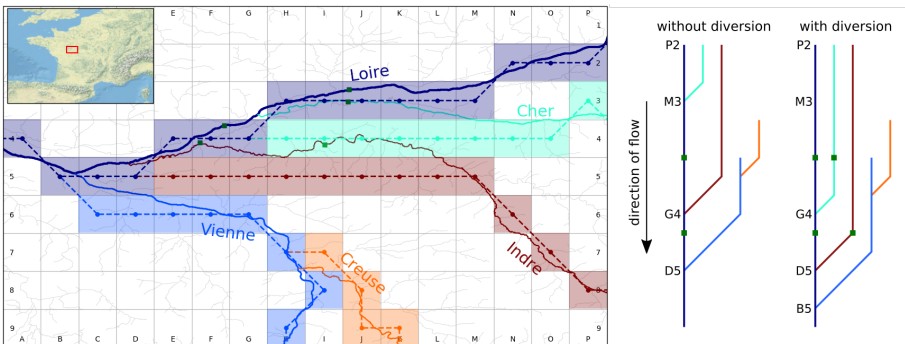

**Figure 3.** Left panel: example of river diversions within the Loire River basin (France); right panel: directional tree of the river network shown in the left panel. As in Fig. 2, rivers are treated in descending order of their drainage area: 1. the Loire river (dark blue), 2. the Vienne river (light blue), 3. the Cher river (green), 4. the Creuse river (orange) and 5. the Indre river (red). Solid lines and dashed lines represent rivers at HR and 12D, respectively. Green squares represent gauge stations.

An example of diversion in the Loire river basin (France) is shown in Fig. 3. The Loire river (dark blue) is the major river of the Loire basin and is treated first. Second is the Vienne river (light blue), followed by the Cher river (green), the Creuse river (orange) and the Indre river (red). The M3-to-H3 cells are occupied by the Loire river at 12D, so that the Cher river portion within these cells has to be diverted to the M4-to-H4 cells. Similarly, the cells within L4-to-E4 are occupied by the Cher river and the Loire river at 12D so that the Indre river is diverted (L5-to-E5). River diversion allows to conserve as much as possible the river network structure, even when several rivers flow within the same cell. Without diversion (e.g., river merging at cell M3), both gauge stations (green squares) in the J3 cell would be associated to the Loire river, while one of them is located in the Cher river. River diversion allows to conserve the location of gauges as well as river nodes (confluences) within the river network tree.

Hypsometry (elevation with respect to the longitudinal distance along the river) is computed during the process so that at the end of the process values of river length, slope and elevation are assigned to each cell. Fig. 4 shows the hypsometry curves of the rivers shown in Fig. 2. Hypsometry is interpolated in case of diversion. In addition, a unique identifier has been assigned to each upscaled river and its corresponding river in the original HR network (as shown by colors in Figs. 2 and 3). This identifier and the hypsometry are used thereafter to derive the hydro-geomorphology parameters.

The final river network at 12D at the global scale is represented in Fig. 5, and zooms over selected regions are proposed in Fig. 6. For comparison purposes, Fig. S1 presents the river network over the same selected regions but at the HD resolution.

The type of river network required by most of river routing models (especially those working with the D8 convention) has to provide a flow direction for each cell of the model. This ensures the closure of the global scale water budget in Earth System Models. The type of soil (nature, river, lake, cities etc.) and other characteristics (such as climate zone) are then not considered to set up the global scale river network. As a consequence, flow directions are also derived over arid regions where no river

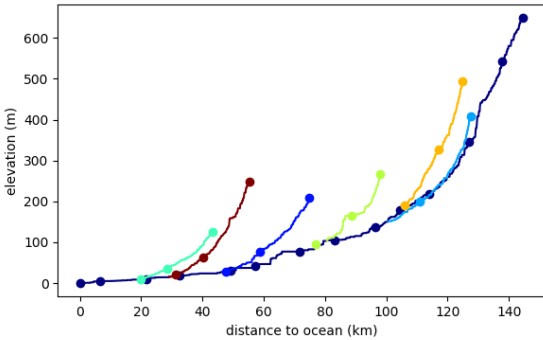

**Figure 4.** Example of hypsometry curves for rivers of Fig. 2.

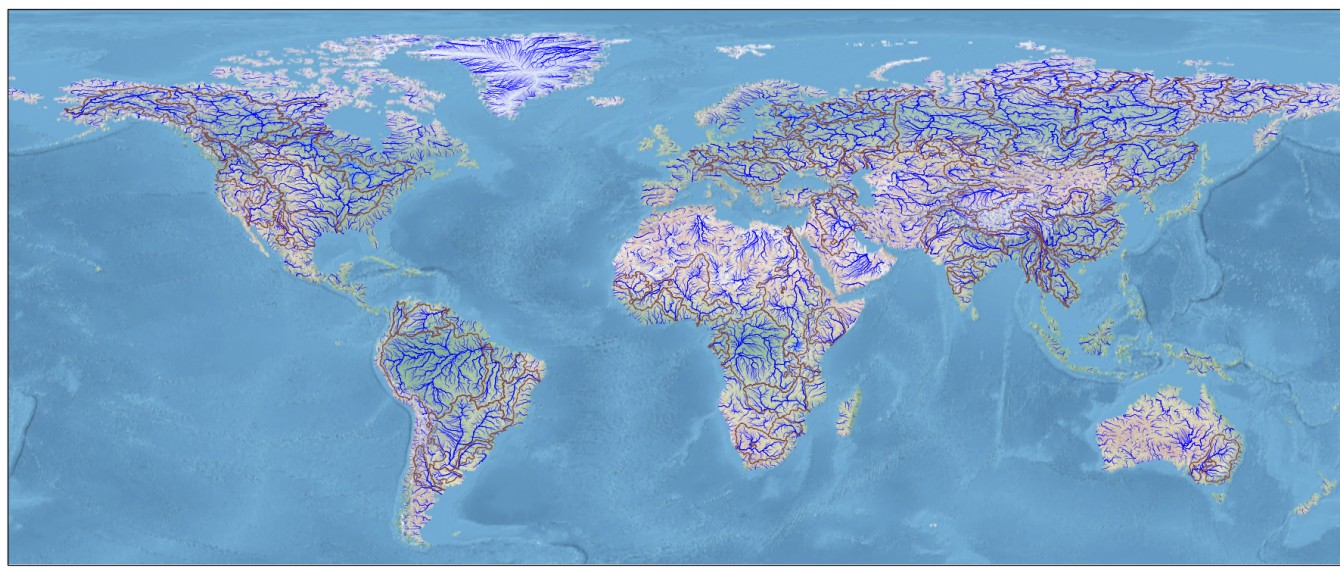

**Figure 5.** Global scale river network at 1/12° resolution. The largest 69 basins of the world used for the quality assessment are delineated in brown.

exists or within cells where no headwater stream has developed. In that sense, the river network should be considered as a drainage network.

## 2.3 Quality assessment

The DRT algorithm as been designed to conserve the river network structure as much as possible. The hierarchical river selection and river diversion have been set up to that purpose. The quality of the resulting river network has been assessed by Wu

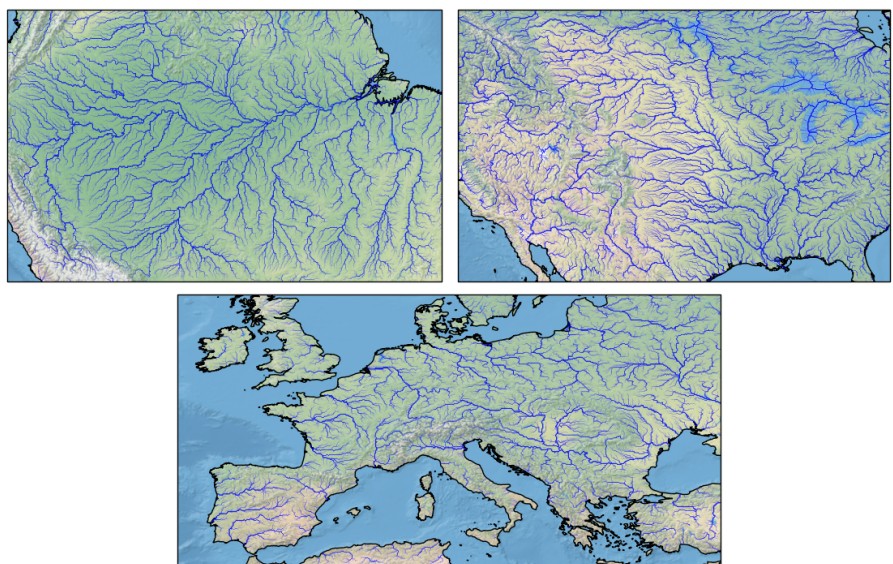

**Figure 6.** Regional scale river network at 1/12° resolution: Amazon basin (top-left), North America (top-right) and Europe (bottom).

et al. (2011, 2012). Here the 69 largest river basins (shown in Fig. 5) have been assessed both qualitatively and quantitatively. Their total area equals $65.10^6$ km$^2$, which represents half of the total land area (excluding Antarctica and Greenland).

The qualitative assessment consists of visual comparisons of river network from different sources, including the original MERIT-Hydro, the previous version of the CTRIP river network (CTRIP-HD) and Google Earth images. The shape of the basins boundary has also been compared with those from CTRIP-HD, the original DRT network at 12D and the GRDC database (Lehner , 2012). For the latter, basin boundaries have been derived from the HydroSHEDS dataset at gauging stations within the largest 405 basins of the world. Basin boundary delineation has been carefully checked and is considered as high quality (Lehner , 2012).

The quantitative assessment first relies on the relative difference between the basin area from the newly developed 12D river network and from other datasets, including the original DRT, MERIT-Hydro and GRDC. In addition, to assess the basin shape and coverage, an Intersection-over-Union (IoU) index is computed as:

$$\text{IoU} = 1 - \frac{\text{area of the intersection basin mask}}{\text{area of the union basin mask}} \tag{1}$$

The IoU index is applied on the basin masks computed from the new 12D network and the original DRT network. It equals 0 in the perfect case where masks exactly overlap, and reaches 1 when both masks do not intersect. Details of the statistics are gathered in Table S1 in supplementary material.

Over the 69 largest basins, the overall agreement between MERIT-Hydro and the 12D river network is very good, with a median relative area difference of 0.3 %, which demonstrates the robustness of the upscaling algorithm. Among the main differences, a large part can be attributed to basins crossing arid regions. When neglecting such basins, the mean relative difference drops from 5.8 % to 3.7 %. This cause of differences is discussed thereafter.

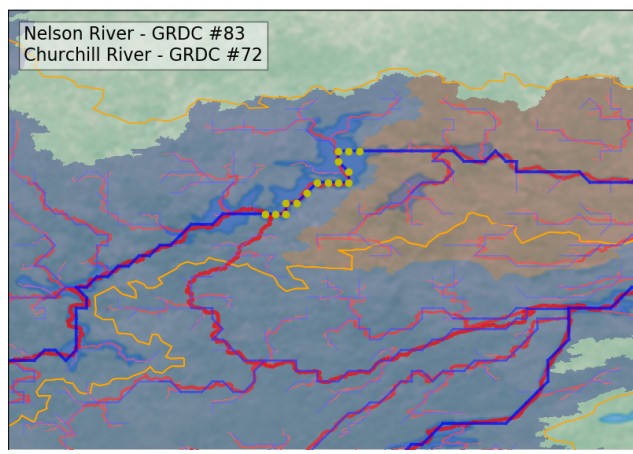

**Figure 7.** Region surrounding the South India Lake in Canada where the river network has been corrected to follow the natural outlet of the lake to the Churchill River. Blues and Red lines represent the river network at 12D and HR (MERIT-Hydro) respectively. The yellow line corresponds to the Nelson River and Churchill River delineation from GRDC. The yellow circles show the cells where the flow direction has been inverted to reconnect the lake to the Churchill River. The blue and red background masks correspond to the Nelson River and Churchill River basins, respectively, extracted from MERIT-Hydro.

Only two other basins are significantly different in the HR and 12D networks: the Nelson River and the Churchill River (Canada). Both river basins are connected via the South Indian Lake. The natural outlet of this lake flows into the Churchill River but the lake is anthropized and a part of the lake volume is diverted to the Nelson River basin for management purposes. The developer of MERIT-Hydro chose the Nelson River to be the major outlet of the South Indian Lake, considering the existing diversion project. We decided to disconnect this outlet, preferring to preserve the natural river network. Fig. 7 zooms over the region surrounding the South Indian Lake. Yellow circles denote cells where the flow direction has been inverted to reconnect the lake to the Churchill River. Another noticeable difference can be shown in the Amur River basin (Asia) in which the Kherlen River appears disconnected to the Argun River, a tributary of the Amur River, while both are connected at Lake Hulun in the GRDC database. Lake Hulun is usually an inland lake without outlet, but in wet periods it may overflow and then join the Argun River (Brutsaert and Sugita , 2008). As for the South Indian Lake, the developer of MERIT-Hydro preferred to keep them separated, which is reflected in the 12D river network (Fig. S2).

When comparing to GRDC and DRT, the averaged relative area difference equals 5.6 % and 8.4 %, respectively. The median reaches 0.8 % in the comparison with DRT. This shows that except for a few basins, the 12D river network and the original DRT are quite close. In Table S1, cells showing a relative area difference higher than 0.10 (10 %) are highlighted, and the potential cause of the difference is indicated by the background colour. Three main causes have been identified.

Most of the differences with GRDC and DRT come from arid conditions characterizing parts of some basins (with a red background in Table S1). In such regions, the terrain is generally quite flat and often disconnected to the river network (endoreic). It is thus quite difficult to extract river networks, which explains the differences between the datasets (as for example

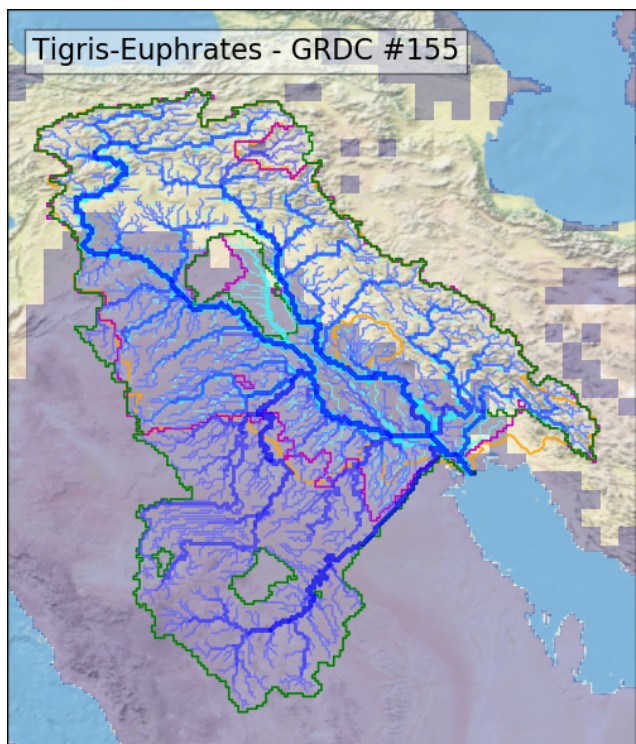

**Figure 8.** Tigris-Euphrates river system. River network from the new algorithm and from DRT is drawn in blue and in cyan, respectively. Basin boundaries from the new algorithm, from DRT and from GRDC and drawn in green, magenta and orange, respectively. The overlapping blue mask represents arid regions. The IoU for this basin equals 14 %, and decreases to 8 % when removing arid regions.

within the basins of Yellow River, Tigris-Euphrates, Senegal, Xi and Rufiji). Nevertheless, the small amount of precipitation that can fall in such regions is partly infiltrated and mostly evaporated. This volume of water never reaches the river network, so differences between river networks over arid regions can be neglected. This can be accounted for in the IoU index by removing arid regions from basin masks, arid regions being defined as regions where the mean annual runoff is below a threshold fixed to 1 mm/yr. Fig. 8 shows that the new 12D river network differs from GRDC in the Southern part of the Tigris-Euphrates river system. Note that DRT is quite similar to GRDC in terms of basin delineation. This major difference can be neglected since it is within the arid region of the Arabian Peninsula. In most of the cases, the IoU significantly decreases (down to less than 10 %) when removing arid regions from the masks for basins showing large differences due to arid regions.

Another source of differences is related to some missing tributaries (green background in Table S1). This is the case for many river deltas, including in the Tocantins, the Xi, the Ural, the Dvina and the Chao Phrava basins. With a D8 convention, models cannot simulate river divergence (a cell can flow into only one other cell). Fig. 9 shows the case of the Red River that joins the Delta of Mississippi, but not in the main branch. This results in different river mouths in MERIT-Hydro and thus in the new 12D network.

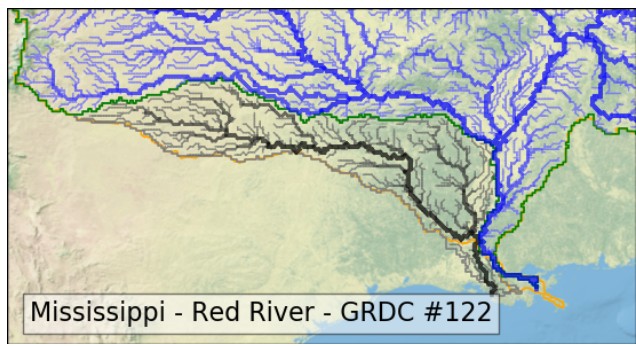

**Figure 9.** Lower Mississippi basin and Red River basin joining in the Mississippi delta. The Mississippi river network is drawn in blue, the Red River in black, while their boundaries are in green and grey, respectively. The orange line represents the basin boundary of the Mississippi river from GRDC.

The last noticeable difference is in the Neva river basin. It appears that in GRDC and DRT, Lake Saimaa (Finland) is disconnected from the Vuoksi river that flows into Lake Ladoga (Russia). As for the South Indian Lake, a significant part of water is derivated from Lake Saimaa to feed canals used for anthropogenic purposes (hydroelectricity, fluvial transport), which may reliably explain the disconnection of this sub-basin.

Finally, the upscaling algorithm produced a reliable and consistent global river network at 12D, very close to the GRDC database in terms of basin delineation for the 69 largest basins of the world. Since MERIT-DEM improved the HydroSHEDS high resolution river network (Yamazaki et al. , 2017), it is expected that the newly developed network improved the original DRT network.

## 3   Derivation of hydro-geomorphology parameters

Large scale river routing models make use of a river network (flow direction) to propagate runoff within river basins to the oceans (in case of exorheic basins). But the propagation dynamics also depends on geomorphological characteristics. These include river geometry (length, slope, width) and roughness (friction coefficient). For models that simulate floodplains, the topography is also needed (generally given as relationships between the surface elevation, the area of the floodplain and the volume of water), as well as the roughness in the floodplains. Similarly, simulating the dynamics of groundwater and the exchanges with rivers, additional parameters are needed, such as soil porosity and transmissivity (or hydraulic conductivity). For large scale models, floodplains and groundwater are usually simulated using a sub-grid approach, for example via a description of the distribution of the topography with respect to the elevation within each cell. This section describes the derivation of river parameters, as well as floodplain and groundwater sub-grid distributions, consistent with the river network derived in the previous section.

### 3.1 River parametrization

A set of parameters related to rivers and describing the flow dynamics within the river network is derived in this subsection.

For each cell, the river slope and bed elevation parameters are directly derived from the elevation of the original MERIT-Hydro adjusted elevation during the upscaling of the river network, by considering the river reach at HR associated to each 12D cell. It should be noted that applying DRT is quite similar to the vector-grid-hybrid method (Yamazaki et al. , 2013) for the extraction of the ground elevation of each cell except that the representative area of each pixel is not computed from the HR sub-catchment but directly from the HR river stretch within the cell. Nevertheless, given the type of model the current dataset is developed for (simplified global scale routing model), and given the uncertainties at this resolution in runoff fields used to force the model, we suppose that the difference in area is negligible, at least for catchments covering several grid cells (or equivalently area greater than 1000 km$^2$).

For the river length within each cell, the computation relies on the HR route within the cell, contrary to other methods that use the flow direction to compute the distance between the center of the cell and the center of the following cell and that multiply this distance by a constant meandering ratio (e.g., CTRIP-HD). Here, meanders are accounted for in the computation of distances in the HR river route. The final river length within each cell is bounded between 1000 m and 20000 m. One may note that river reaches shorter than 1000 m correspond to headwaters, while reaches greater than 20000 m correspond to highly meandering rivers. River slope is also bounded between $10^{-4}$ m/m in flat regions and 0.5 m/m in mountainous areas.

The river width $W_{riv}$ is mainly derived from the Global River Width from Landsat dataset (GRWL, Allen and Pavelsky , 2018; Frasson et al. , 2019). GRWL was developed by processing Landsat imagery at approximately mean annual flow. It provides high-resolution centerline locations alongside river width for global rivers wider than 30 m. Water body type is given for each river reach as metadata. Here, reaches corresponding to lake or reservoir, canal, or tidally influenced river are discarded. Since the location of river centerlines may not match exactly the river network at 12D, river centerlines have first been clipped on the MERIT-Hydro river network. Then the river identifier making the correspondence between HR and 12D is used to derive the river width for each cell at 12D (based on the median of HR river width within each 12D cell). For grid cells where no river width can be derived from GRWL, we used the empirical relationship developed for previous versions of the CTRIP model (Vergnes et al. , 2014), based on the annual mean discharge $Q_{mean}$:

$$W_{riv} = 5.41 Q_{mean}^{0.59} \qquad (2)$$

$Q_{mean}$ is derived from the runoff field in the GG-HYDRO database (Cogley , 2003) propagated through the river network. A threshold of 30 m is chosen as the minimum width. Figure 10 presents the distribution of river width from GRWL with respect to the annual mean discharge. It shows a strong relationship that is well captured by the empirical relationship from Vergnes et al. (2014).

Finally, a smoothing is applied over each river (moving average with a 16-pixel width) to avoid unrealistic shrinkages (see Fig. 11).

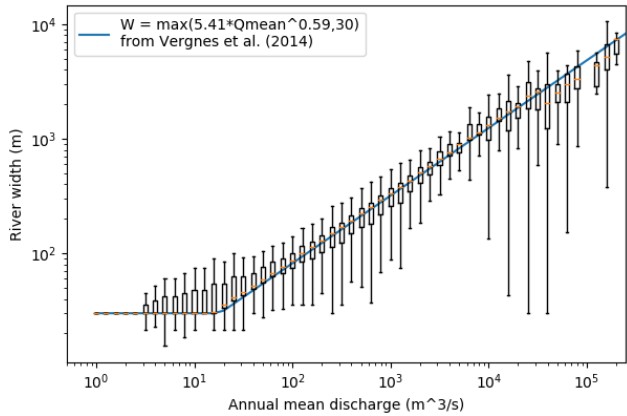

**Figure 10.** Distribution of river width from GRWL (Allen and Pavelsky , 2018) with respect to the annual mean discharge. The blue solid line represents the river width derived from the empirical relationship proposed by Vergnes et al. (2014).

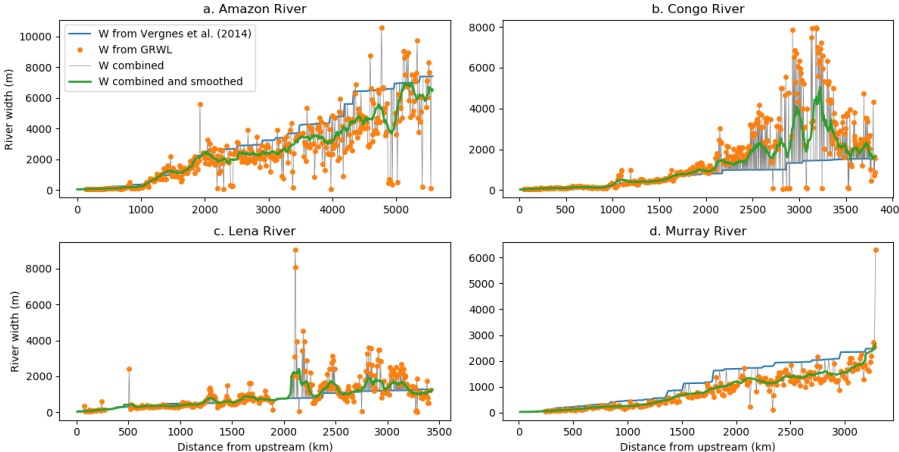

**Figure 11.** Examples of combination of river widths from GRWL and Eq. (2) for (a) the Amazon River, (b) the Congo River, (c) the Lena River and (d) the Murray River.

The river depth $h_{riv}$ (or bankfull depth), used to simulate floodplains, is derived from Eq. (3), as in Decharme et al. (2019):

$$h_{riv} = 1.4W_{riv}^{0.28} \tag{3}$$

The last parameter related to river hydro-geomorphology is the roughness coefficient. Here we used the same methodology as in Decharme et al. (2019). The roughness coefficient $n_{riv}$ is derived from a weighted geometrical average between a value of 0.035 s.m$^{-1/3}$ (a standard value for quite large rivers, Lucas-Picher et al. (2003); Yamazaki et al. (2011)) and a roughness coefficient $n_{fld}$ describing the riparian zone and the vegetation in the potentially surrounding flooded area (described in the

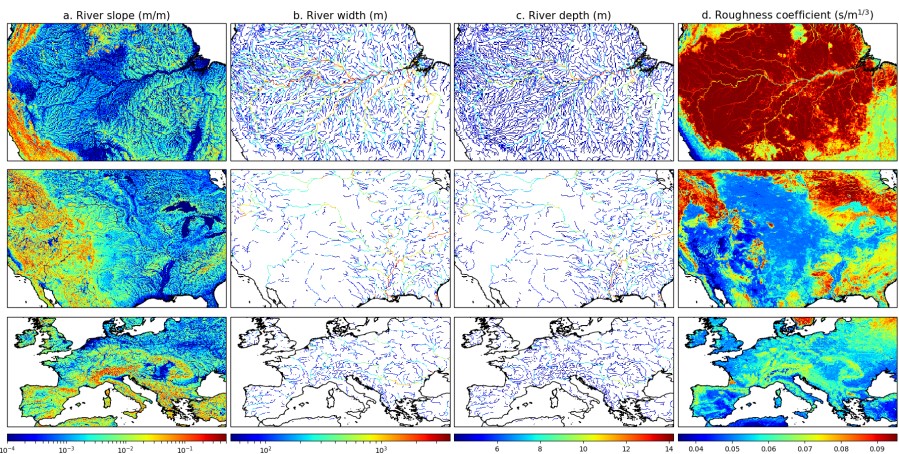

**Figure 12.** River parameters over Amazon (first row), USA (second row) and Europe (third row): river slope (a), river width (b), river depth (c) and roughness coefficient (d). River width smaller than 50 m and river depth smaller than 4 m have been filtered out for clarity.

next section):

$$n_{riv} = 0.035^{1-\alpha_r} \times n_{fld}^{\alpha_r} \tag{4}$$

The weighting coefficient $\alpha_r$ varies linearly from 1 in the headwater cells to 0.5 at the outlet of the river basin:

$$\alpha_r = \frac{1}{2} \left( \frac{SO_{max} - SO}{SO_{max} - SO_{min}} + 1 \right) \tag{5}$$

300  where $SO$ is the stream order within the river basin, $SO_{min}$ and $SO_{max}$ are the minimum and maximum stream order values within the same basin.

Figure 12 shows the different parameters over different regions of the globe (Amazon basin, USA and Europe).

## 3.2  Floodplains parametrization

Floods may occur when the water height within the river exceeds the river depth, then causing lateral flows over the river
305  banks. Floodplains, described as the area surrounding the river which can be flooded during heavy rain events, acts as a water storage and directly impacts the water propagation within the river network. High accuracy representation of flow dynamics within floodplains requires a highly accurate DEM and intense computations to solve the 2D Saint-Venant equations. This can be done over small areas, such as urban areas, but not at regional scales. A number of large scale river models which account for floodplains and their impacts on the flow dynamics are based on subgrid approximations (Yamazaki et al. , 2011; Decharme
310  et al. , 2012). The concept generally relies on computing the volume of water that flows outside the river (given the river maximum volume) and estimating the water level and the area of the flooded zone from subgrid distributions (Decharme et al. , 2012).

**Table 1.** The 12 land types derived from the 1-km ECOCLIMAP-II database and their corresponding Manning roughness values.

| Number | Land type | $n_i$ |
|---|---|---|
| 1 | Bare Soil and Desert | 0.035 |
| 2 | Rocks and Urban area | 0.035 |
| 3 | Permanent Snow and Ice | 0.035 |
| 4 | Temperate Broadleaf Deciduous | 0.075 |
|   | Tropical Broadleaf Deciduous | |
|   | Temperate Broadleaf Evergreen | |
|   | Boreal Broadleaf Deciduous | |
|   | Shrub | |
| 5 | Boreal Needleleaf Evergreen | 0.100 |
|   | Temperate Needleleaf Evergreen | |
|   | Boreal Needleleaf Deciduous | |
| 6 | Tropical Broadleaf Evergreen | 0.100 |
| 7 | C3 crops | 0.050 |
| 8 | C4 crops | 0.050 |
| 9 | Irrigated crops | 0.050 |
| 10 | C3 Grassland | 0.050 |
|   | Boreal Grassland (Tundra) | |
| 11 | C4 grassland | 0.075 |
| 12 | Peat, bogs and Irrigated grass 0.5-1.0 | 0.075 |

Here, in order to ensure the consistency between the river network and the floodplain representation, the adjusted elevation from MERIT-Hydro is used as the baseline to compute the subgrid distributions of elevation, cell fraction (related to the area) and volume of water within the floodplain. The method to extract these distributions is described in Decharme et al. (2012).

Floodplain roughness to estimate the flow velocity between the river and the floodplain, using the Manning-Strickler equation. In addition, a floodplain roughness coefficient is estimated empirically as in Decharme et al. (2019). This coefficient is directly related to the type of land within the cell. The ECOCLIMAP-II land cover database (Faroux et al. , 2013) was used to characterize the type of land. For each cell at 1/12°, we computed the fraction $f_i$ of each land type in Table 1. Then the floodplain roughness was computed as the weighted average of default values $n_i$ for each land type as given in Table 1:

$$n_{fld} = \sum_{i=1}^{12} (f_i \times n_i) \tag{6}$$

### 3.3 Groundwater parametrization

As floodplains, aquifers can significantly impact the propagation of water within rivers. Aquifers are usually recharged by the infiltration of water at the surface and can interact directly with rivers. The direction of the exchanges between rivers and aquifers depends on the water elevation in the river and the water table depth. As for floodplains, some large scale hydrology models (e.g. Döll and Fiedler , 2008; Vergnes and Decharme , 2012; Decharme et al. , 2019) integrate a simplified representation of aquifers in order to better represent the continental water cycle, and more specifically the water propagation within the river network.

To delineate the main aquifers that could be represented in large scale hydrology models, Vergnes and Decharme (2012) used the global map from the Worldwide Hydrogeological Mapping and Assessment Programme (WHYMAP; http://www.whymap.org). As in Vergnes and Decharme (2012), we considered two of the three categories included in this map, for which the two-dimensional diffusive solver is well adapted: the "major groundwater basin" that gathers sedimentary basins and the alluvial plains with permeable materials, and the "complex hydrogeological structure" which includes (among others) alluvial aquifers formed by the deposition of weathered materials. The "local and shallow aquifers" category corresponds to the old geological platforms characterized by crystalline rocks with scattered, superficial aquifers, and is not considered here. Finally, mountainous cells are removed by using a criteria on terrain slopes and the global lithology map from Dürr et al. (2005) was used to refine the delineation of aquifers. Examples of aquifer delineation are shown in Fig. 13(a).

In Vergnes and Decharme (2012), the groundwater dynamics is described by a two-dimensional diffusive equation which requires some additional parameters characterizing the soil, such as the effective porosity and the transmissivity. These characteristics highly depends on the lithology and can be estimated by mean values from the literature. Here, the lithology was derived from Dürr et al. (2005) and the mean values from Table 1 in Vergnes and Decharme (2012). Note that values of porosity and transmissivity have been capped at 0.05 $m^3/m^3$ and 0.02 $m^2/s$, respectively, in order to avoid a too high inertia within the corresponding aquifers. Values of both parameters are shown in Fig. 13(b-c) over different regions of the globe.

Lastly, to simulate the exchanges between aquifers and rivers, the piezometric head has to be simulated and compared to the water level within the river. The piezometric head may be also used to represent upward capillary fluxes up to the vegetation root layer (Vergnes et al. , 2014). As for floodplains, a subgrid approach may be used, as in Vergnes et al. (2014); Decharme et al. (2019), to derive the distribution of the elevation and the cell fraction within each cell. Here again, the adjusted elevation from MERIT-Hydro is used to compute these distributions.

## 4 Validation

### 4.1 Modelling configuration

In this section, we set up a model configuration with the river network and the parameters described in sections 2 and 3. For this validation step, the CTRIP model is chosen with the same configuration as in Decharme et al. (2019). In the latter study, CTRIP is operated at 0.5° resolution (CTRIP-HD) and the groundwater and floodplain components are accounted for. CTRIP-

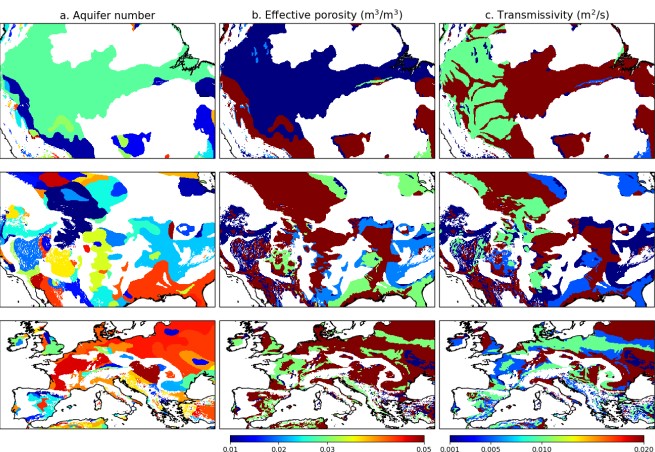

**Figure 13.** Aquifers numbering and parameters over Amazon (first row), USA (second row) and Europe (third row): aquifer number (a), effective porosity (b) and transmissivity (c).

HD has been extensively validated against various types of observations, including river discharge, flood extent, groundwater head, total water storage (Alkama et al. , 2010; Decharme et al. , 2012, 2019; Vergnes and Decharme , 2012; Vergnes et al. , 2014). The whole set of hydro-geomorphological parameters derived in this paper at 12D are available for CTRIP-HD. Consequently, the new set of parameters can be evaluated and compared to its lower resolution while keeping a consistent modelling framework.

Both configurations are forced by runoff and drainage fields generated by the ISBA land surface model as described in Decharme et al. (2019). The atmospheric forcings to ISBA are the Earth2Observe (E2O) dataset. Although the ISBA and CTRIP models are fully coupled in Decharme et al. (2019), we prefer here to run the CTRIP model in offline mode; then the configuration considered here includes the representation of floodplains and aquifers, but backward fluxes to ISBA (capillary rise and evaporation over floodplains) are neglected. The half-degree runoff and drainage fields are downscaled at 12D with a simple nearest neighbour method and provided to the CTRIP model at a daily time step over the period 1979-2014. The CTRIP simulation time step is set at 3600 s and the time step for output river discharge is 24 h (daily). Finally, a 30-year spinup period was used to let the groundwater storage state variable reach its equilibrium value.

## 4.2 Evaluation strategy

Here we compare the performances of the new configuration (CTRIP-12D) to those of the previous one (CTRIP-HD). The performances mainly relies on comparisons between simulated and observed discharge for more than 10,000 *in situ* gauge stations over the globe.

**Table 2.** Description of the databases considered for the selection of *in situ* gauges stations with at least 3 years of discharge observations within 1979-2014. *Last access to all websites in 25-02-2021.*

| Database | Region | Stations | Reference |
|---|---|---|---|
| Global Runoff Data Centre | Globe | 4769 | http://www.bafg.de/GRDC/EN/Home/homepage_node.html |
| USGS | United States | 5205 | http://waterdata.usgs.gov/nwis/sw |
| HYDAT | Canada | 1652 | https://collaboration.cmc.ec.gc.ca/cmc/hydrometrics/www/ |
| French Hydro database | France | 914 | http://www.eaufrance.fr |
| Spanish Hydro database | Spain | 492 | http://ceh-flumen64.cedex.es/anuarioaforos/default.asp |
| HidroWeb | Brazil | 270 | http://www.snirh.gov.br/hidroweb/ |
| R-ArcticNet | Northern Asia | 133 | http://www.r-arcticnet.sr.unh.edu/v4.0/AllData/index.html |
| China Hydrology Data Project | China | 67 | Henck et al. (2011) |
| HyBAm | Amazon basin | 14 | https://hybam.obs-mip.fr/ |

### 4.2.1 River discharge datasets

A large number of *in situ* gauge stations have been considered for the comparison with simulated discharge. The data was extracted from various open access databases described in Table 2. A minimum of 3 years of records over 1979-2014 was imposed as a mandatory criterion, as well as the presence of localization and drainage area in the station metadata. A total of
13,516 stations was finally selected with drainage area ranging from 400 km$^2$ to 4.7 10$^6$ km$^2$.

### 4.2.2 Localization of gauge stations

For the comparison between observed and simulated discharges, one must first localize the gauge station within the river network of the model. A very common method consists of looking for the grid pixel surrounding the station, for which the drainage area is the closest to the one reported in the station metadata. Yet, in some cases, this can lead to an erroneous selection
of the CTRIP pixel corresponding to a certain station. Such problems can happen fortuitously (see the example in Fig. 14) or for portions of rivers that have been diverted during the generation process (section 2). This highlights the necessity to improve the localization methodology.

Since the coordinates and the drainage area of each station is known, it is possible to delineate the catchment related to the station from the MERIT-Hydro database. First the pixel in the HR grid corresponding to the station is first designated by the
one minimizing a criterion that combine the distance to the station and the drainage area. At such a high resolution, the method can be considered robust enough to avoid mislocalization.

The second step consists of sorting the CTRIP pixels around the station (as in Fig. 14) by descending drainage area. For each pixel the comparison between the catchment delineation obtained from MERIT-Hydro and from CTRIP is quantified by

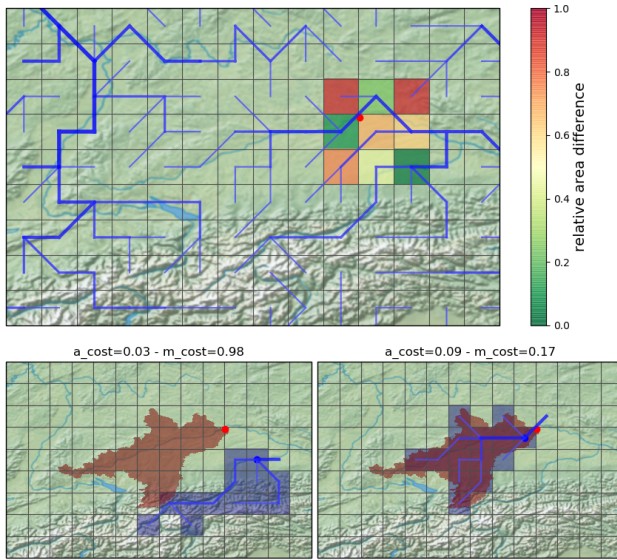

**Figure 14.** Example of necessary relocalization using the mask overlapping method. The station (red dot) is the Oberndorf station (GRDC id 6342910) over the Danube river (48.947°N, 12.0149°E). In the two lower panels, the real basin (from MERIT-Hydro) is shown in red. The lower left panel shows the CTRIP basin in blue for the cell with the drainage area the closest to the drainage area reported at the station (the area relative error $a_{cost}$ is the lowest). The lower right panel shows the CTRIP basin for the cell with the lowest mask overlapping relative error ($m_{cost}$).

computing the IoU index (Eq. (1)). The area relative error ($a_{cost}$) and the mask overlapping relative error ($m_{cost}$) are finally
combined to find the best candidate.

Consequently, each station is assigned a CTRIP pixel more consistently than when using classical approaches. This process is applied for CTRIP-HD and CTRIP-12D. It also ensures that basins smaller than one grid pixel are excluded from the selection since $m_{cost}$ would be too high. Note also that the method is able to solve potential localization difficulties due to the river diversion allowed during the network generation process (section 2). Although the river diversion can foster this kind of
situation, it allows in the same time to correctly localize the confluences within the network. This avoids artificial confluences and would consequently prevent the stations concerned to be discarded (due to a bad mask overlap).

### 4.2.3   Evaluation metrics

The main metrics used to quantify the performances of each simulation is the modified Kling-Guppta-Efficiency (KGE, Kling et al. , 2012). The KGE is a combination of three factors describing the error in terms of relative bias, correlation and relative

variability (Eq. (7)). KGE varies from $-\infty$ to 1, the upper bound corresponding to simulation results that perfectly match the observations.

$$\text{KGE} = 1 - \sqrt{(r-1)^2 + (\beta-1)^2 + (\gamma-1)^2} \tag{7}$$

$$\beta = \frac{\mu_s}{\mu_o}$$

$$\gamma = \frac{\sigma_s/\mu_s}{\sigma_o/\mu_o}$$

where $r$ is the correlation coefficient between simulated and observed discharge, $\beta$ is the bias ratio, $\gamma$ is the variability ratio, $\mu$ is the mean discharge and $\sigma$ the standard deviation.

We also use the Normalized Information Contribution (NIC) particularly suited to quantify the improvement between two simulations, as in Albergel et al. (2018):

$$\text{NIC} = \frac{\text{KGE}_{\text{new}} - \text{KGE}_{\text{ref}}}{1 - \text{KGE}_{\text{ref}}} \tag{8}$$

where $\text{KGE}_{\text{ref}}$ is the KGE criterion for the reference simulation and $\text{KGE}_{\text{new}}$ is the KGE criterion for the simulation which is compared to the reference. The advantage of the NIC criterion is that it normalizes the difference between the KGE of two experiments. A given KGE difference has not the same impact in terms of performance depending on the value of KGEs. For instance, if $\text{KGE}_{\text{ref}}$=0 and $\text{KGE}_{\text{new}}$=0.2 then NIC=0.2, whereas if $\text{KGE}_{\text{ref}}$=0.8 and $\text{KGE}_{\text{new}}$=1 then NIC=1. The higher NIC

value in the second case means that the improvement is better (perfect in that case) although the difference is the same.

## 4.3   Simulation performances

### 4.3.1   Evaluation of CTRIP-12D

In this section, the modelling results are evaluated by comparing simulated and observed river discharge at the 13,516 gauge stations selected from various open access databases as described in section 4.2.1.

Figure 15 shows the KGE value for the 11,238 stations with a KGE greater than -1 (the others have been discarded in this figure for a sake of clarity), and zooms over the Amazon basin, North America and Europe. Globally, the CTRIP-12D model clearly shows quite good performances, whatever the basin area, especially in South America, Europe, South-East Asia and Eastern part of USA. The KGE decreases when the CTRIP-12D model is unable to satisfactorily reproduce river discharge. Among all the stations, 2,278 show a KGE lower than -1 which correspond to very poor performances. Nevertheless, it has to

be noted that most of these stations have a quite small drainage area (90 % have a drainage area smaller than 50,000 km$^2$, 75 % smaller than 10,000 km$^2$). Different reasons are identified to explain such deficiencies. First, some rivers are highly regulated, which is not accounted for here. Second, in some regions the ISBA land surface model may fail to produce realistic runoff used to feed CTRIP, because of model and atmospheric uncertainties (as, e.g., in mountainous areas). Third, in arid regions, the evaporation over open waters (rivers and floodplains) can be very high and is not accounted for here, as for example in

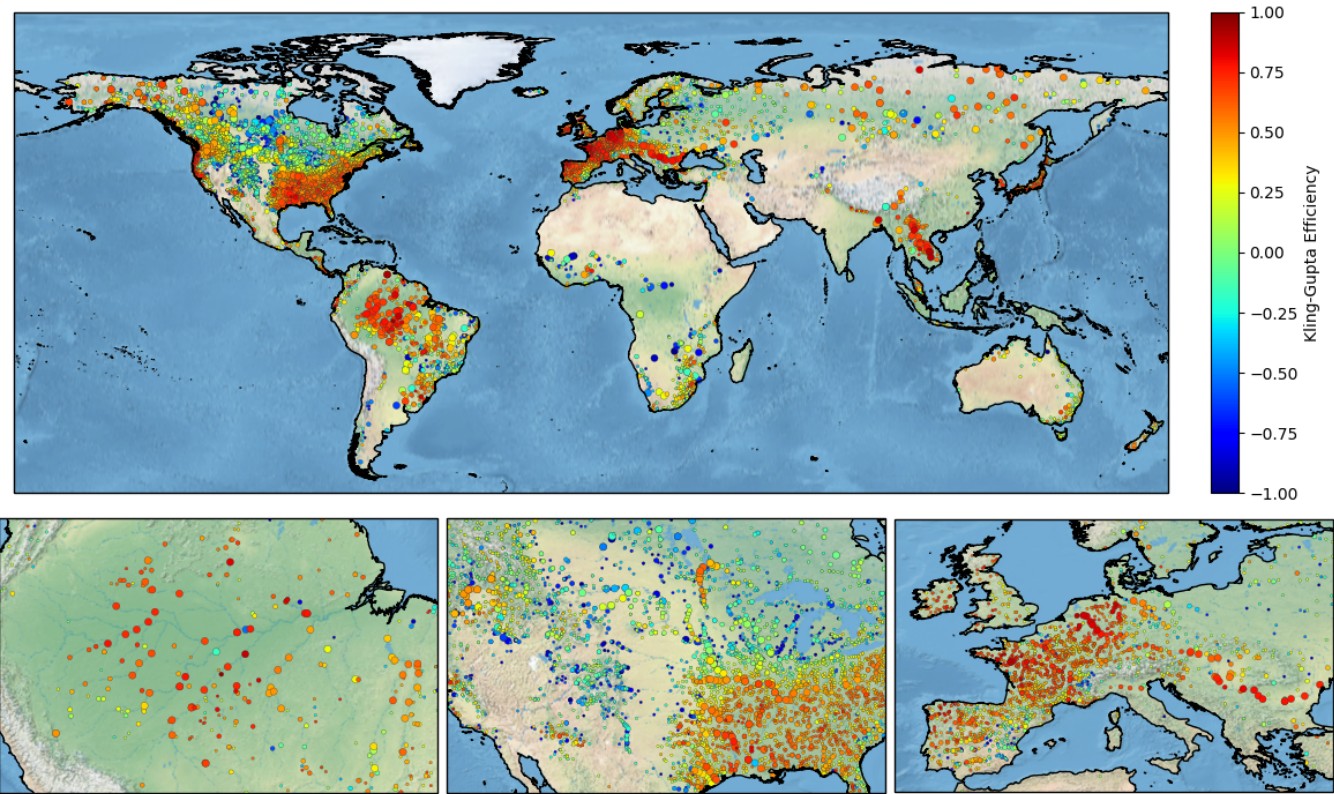

**Figure 15.** Kling-Gupta Efficiency for CTRIP-12D over 11,238 gauge stations (with KGE>-1) and zooms over the Amazon basin, North America and Europe. The size of circles depends on the drainage area at each station.

the Niger basin, which highly impacts the discharge ratio. Finally, the dynamics of lakes is neglected, which also impacts the quality of the results, mainly in terms of correlation and standard deviation.

To verify that poor performances are mainly due to these reasons and not to the new parametrization at 12D, the next section compares the performances of CTRIP-12D with those of CTRIP-HD, both of which being ran in the same configuration.

### 4.3.2 Comparison with CTRIP-HD

Considering that the CTRIP-HD model in its current version has been extensively validated in the past (e.g., Alkama et al. , 2010; Decharme et al. , 2012; Vergnes and Decharme , 2012; Vergnes et al. , 2014; Decharme et al. , 2019), we here mainly focus on the comparison between this existing version of CTRIP and the new one at 12D developed in this article.

By applying the methodology to localize gauge stations within the river network (see section 4.2.2), 2,612 stations have been selected to have a correct localization in the river networks of both CTRIP-HD and CTRIP-12D. For these stations, we

computed the KGE values for both simulations as well as the NIC criterion that quantifies the improvement or degradation of CTRIP-12D compared to CTRIP-HD. As written in the previous section, despite the overall good quality of the CTRIP model,

it may fail in reproducing observed discharges, in particular for stations highly influenced by human activities which are not represented in CTRIP. For these stations, we consider that the CTRIP model is not adapted due to processes not accounted for. Consequently, we consider that improvement or degradation of model performances are not relevant and we discarded these stations. Fig. 16 presents the NIC values for all the stations with KGE values greater than -1 (2,164 stations). It shows that performances are globally better with the new resolution. More precisely, 1,988 stations (92 %) are impacted by the new parametrization ($|NIC| > 0.02$), including river routing (river network and parameters), floodplains (roughness and sub-grid topography) and groundwater (aquifer parameters and sub-grid topography). Among them, 470 stations (24 %) are negatively impacted, while 1,518 (76 %) are positively impacted.

To get a closer look into the differences between performances of CTRIP-HD and CTRIP-12D, panels in Fig. 17 show the distributions of KGE, correlations and $\gamma$ variability coefficient for both simulations. For each criterion, the left panel (a, c or e) shows the different criteria with respect to the drainage area at the gauge stations. For this comparison, no station with a drainage area smaller than 1,000 km$^2$ has been selected because of the low resolution of CTRIP-HD. Whatever the resolution, KGE and correlation increase with the drainage area, but for both criteria, performances are clearly better for CTRIP-12D for all category of drainage area. Similar result is shown with the relative variability depicted by $\gamma$ in Fig. 17(e) with median and quartile values closer to 1 for CTRIP-12D. Right panels (b, d and f) of Fig. 17 also show overall better performances for CTRIP-12D with KGE, correlation and $\gamma$ distributions closer to 1.

Better performances could be expected for smaller basins since these basins are represented by just a few cells at HD, and the difference between the basin delineation at HD and 12D could be relatively high, then leading to different contributing areas. The better performances of CTRIP-12D for larger basins is less expected. Indeed processes and forcing are the same for both configurations and parameters are derived using similar strategies and relationships. The improvement of the correlation and variability demonstrates that a better defined river network improves the dynamics of river propagation within the basin and interactions with floodplains and aquifers. Other potential sources of differences between both models include: 1. the reference HR dataset (HydroSHEDS for CTRIP-HD, MERIT-Hydro for CTRIP-12D), which impacts the generation of floodplains and aquifers sub-grid parametrization; 2. the use of observed-based river width for CTRIP-12D.

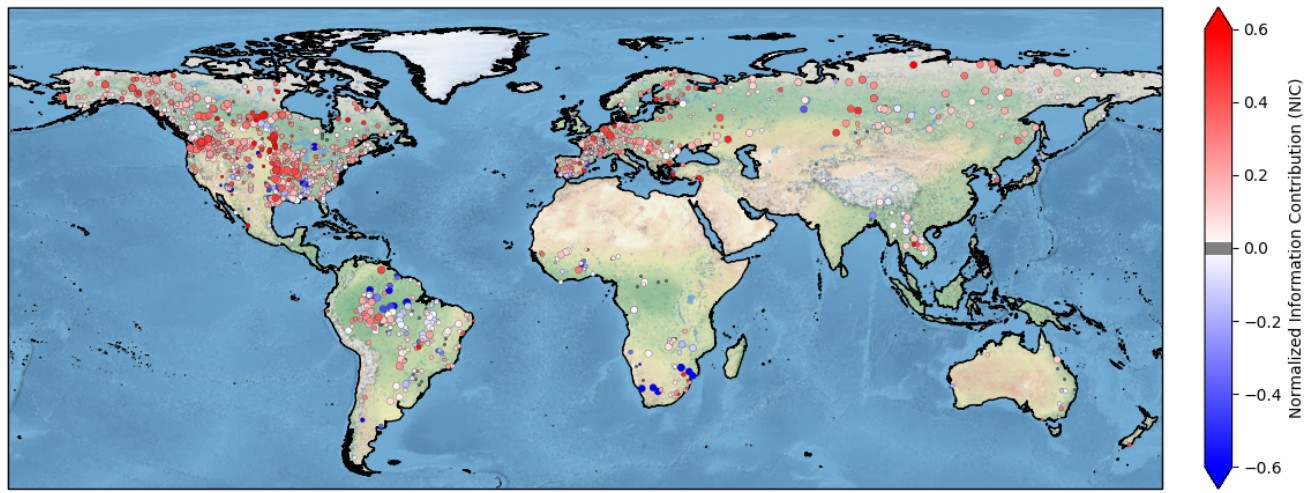

**Figure 16.** NIC of Kling-Gupta Efficiency between CTRIP-12D and CTRIP-HD over 2,164 gauge stations (with KGE>-1). The size of circles depends on the drainage area at each station.

## 5  Conclusions

This article presents a new global scale river network at 1/12° (12D) derived from the MERIT-Hydro high resolution hydrography data. We also provide a set a hydro-geomorphological parameters that are consistent with this new river network. The set of parameters includes: length, width, depth and roughness for rivers, roughness and sub-grid topography for floodplains,
transmissivity, effective porosity and sub-grid topography for aquifers.

The new river network and hydro-geomorphological parameters have been implemented in a new version of the CTRIP model (Decharme et al. , 2019) and assessed through a comparison of simulation performances with the previous version of CTRIP at 1/2° (HD). It is shown that river discharge are overall better estimated with the 12D version and that the improvement can be mainly attributed to the finer representation of the real river network. When increasing the resolution of CTRIP from
HD to 12D, the total number of cells changes from $62 \ 10^3$ to $2.2 \ 10^6$, the total number of basins increases from 4,800 to 56,500 and the total river length increases from $2.5 \ 10^6$ km to $21 \ 10^6$ km.

As a perspective, it can be mentioned that the derivation of some parameters could be improved over some regions by using existing local or national data. For example, aquifers could be better described by the Référentiel Hydrogéologique Français (BDRHF) database available over France, or by hydrogeological maps from USGS over the United States.
In grid-based approaches, the river network is discretized on a regular Cartesian grid, so that unit-catchments are rectangular pixels with their own hydrogeomorphological characteristics. The complete dataset described here is particularly well suited to a number of large-scale RRMs using a gridded structure for global hydrological studies (see Table 2 in Kauffeldt et al. , 2016). Not all of them are currently running at 12D resolution, while, on the other hand, the current tendency suggests that 5 arcmin could become the next standard resolution for global scale climate studies, namely via the recent release of the last global

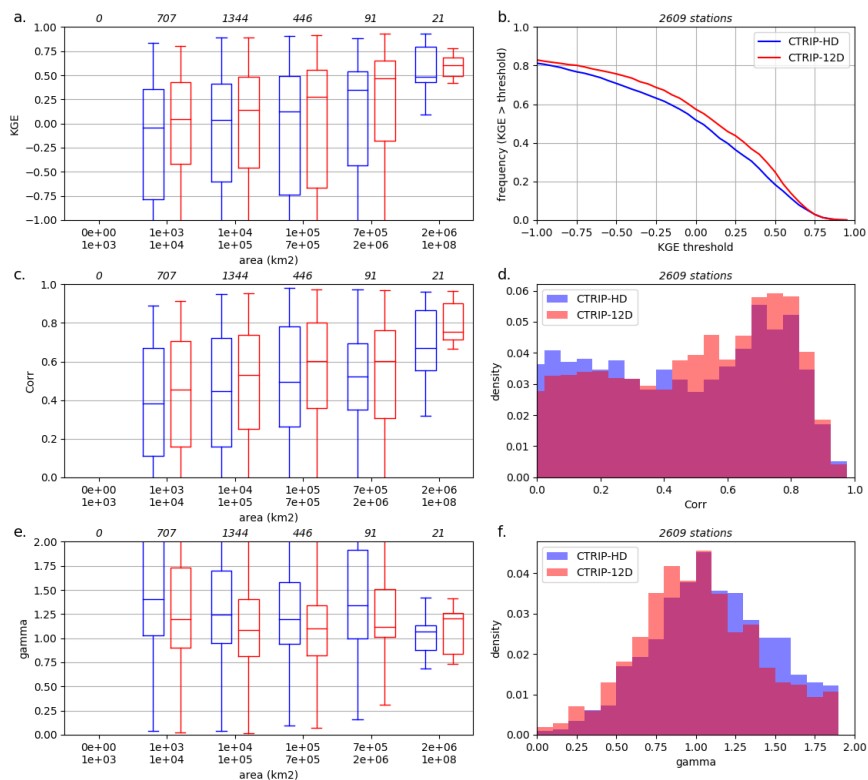

**Figure 17.** Left panels: distribution of KGE (a), correlation $r$ (c) and $\gamma$ coefficient (e) with respect to the drainage area at each station for both CTRIP-HD and CTRIP-12D (numbers above the boxes represent the number of stations within each area bin). Right panels: Cumulative Density Function of KGE (b) and Probability Density Function of correlation $r$ (d) and $\gamma$ coefficient (f).

meteorological dataset for impact models in phase 3a of the Inter-Sectoral Impact Model Intercomparison Project (ISIMIP3a Dirk et al. , 2022). With the entire dataset described here (flow direction, river length, river slope, river bank-full depth, river roughness, floodplains roughness, major groundwater basins boundaries, aquifer transmissivity, and aquifer effective porosity), many hydrological models could improve their river routing module by increasing the spatial resolution. Moreover, this consistent and comprehensive dataset can help modellers to integrate some important processes (such as inundation and 490 groundwater) that are still neglected in some models.

## 6 Code and data availability

The river network and the hydro-geomorphology (including floodplains and aquifers parametrizations) data sets are freely available for download from Zenodo (https://doi.org/10.5281/zenodo.6482906 Munier and Decharme , 2021). The source code is also available in this repository.

*Author contributions.* S. Munier and B. Decharme both designed the study and contributed to the manuscript.

*Competing interests.* The contact author has declared that neither they nor their co-authors have any competing interests.

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
