# Peer review of "River network and hydro-geomorphological parameters at 1/12° resolution for global hydrological and climate studies"

_Earth System Science Data, 2021_

## Author Comment (AC2)

essd-2021-434

**River network and hydro-geomorphology parametrization for global river routing modelling at 1/12° resolution**

Simon Munier and Bertrand Decharme

Author response to reviewer #2

*Reviewer comments are in italic and blue font.*

*This manuscript describes about the new river network data for river routing models. The new data is developed using the latest river topography dataset MERIT Hydro, and its accuracy is assessed using various river-related datasets as well as model simulations using CTRIP. I think the manuscript contains adequate description as a data paper, and the estimated accuracy is promising. Given that the river network map is a widely used fundamental information in many hydrology and earth system science studies, I think the manuscript is worth publishing on ESSD, after minor corrections on a few ambiguous parts.*

We would like to thank the reviewer, Dr Dai Yamazaki, for his valuable comments on the manuscript. We also think that not only the river network map could be useful for hydrology and earth system science studies, but also all the associated hydro-geomorphological characteristics since they are consistent with the derived river network, which could hardly be the case if they are provided by different sources. Bellow are the responses to all the comments raised by the reviewer.

*L68: "some recent studies provide new upscaled river network based on MERIT-Hydro (see, e.g., Eilander et al. , 2021), they do not necessarily follow a D8 convention, and they do not provide model parameters consistent with the new river network (such as sub-grid topography)."*
*Please carefully review the paper Eilander et al. , 2021. Their IHU method also generated D8 format river network, and it also provide some sub-grid topography info. Thus, some descriptions in this sentence is not correct.*

We thank the reviewer for this relevant comment. Indeed, Eilander et al. (2021) derived river network in the D8 format based on MERIT-Hydro using a somehow different upscaling method (IHU). They also provided river length and slope sub-grid parameters consistent with the upscaled river network since they are derived from MERIT-Hydro during the upscaling process. So we agree that our sentence is not correct. It has been rewritten as:
"Although some recent studies provide new upscaled river network based on MERIT-Hydro (see, e.g., Eilander et al., 2021), only a limited set of hydro-geomorphology parameters consistent with the new river network have been derived (such as sub-grid river length and slope)."

*L191: "The automatic algorithm of MERIT-Hydro chose the outlet that flows into the Nelson River basin."*
*Development of MERIT Hydro was done with extensive quality assessment, and some input data such as water mask and elevations are modified to ensure realistic river network. Thus, it is not proper to say "Nelson river is chosen as mainstem by "automatic algorithm". Rather than that, the developer of MERIT Hydro decided that Nelson River to be the major outlet of the South Indian Lake, considering the existing diversion project.*

We agree that this sentence is confusing. Indeed, in the case of the South Indian Lake, a significant part of the water volume is diverted from its original path, the Churchill River, to the Nelson River for water management purposes. Simplified river networks, such as those following the D8 format, are not able to represent such diversions, and it is to the developer to decide whether to conserve the natural flow path or not. Whatever this decision, the performance of river routing model will be limited on both basins if river diversion is significant. More complex models will have to be developed to handle this kind of situation. The sentence has been rephrased as suggested by the reviewer:

"The developer of MERIT-Hydro chose the Nelson River to be the major outlet of the South Indian Lake, considering the existing diversion project."

*L203 Fig. 8*
*Probably it is better to explain that how to treat Lake Hulun is the source of the difference. Lake Hulun is usually an inland lake without outlet, but it is connected to the Amur in flooding year. Whether to include Lake Hulun in Amur basin or not highly depends on the developer's decision.*

The reviewer is right, the main source of the difference between the river networks is the way Lake Hulun is treated. The following sentences have been added at the end of the previous paragraph, and the cause of the differences in Table S1 has been modified accordingly:

"Another noticeable difference can be shown in the upper Amur River basin (Asia) in which the Kherlen River appears disconnected to the Argun River, a tributary of the Amur River, while both are connected at Lake Hulun in the GRDC database. Lake Hulun is usually an inland lake without outlet, but in wet periods it may overflow and then join the Argun River (Brutsart and Sugita, 2008). As for the South Indian Lake, the developer of MERIT-Hydro preferred to keep them separated, which is reflected in the 12D river network."

Brutsaert, W., and Sugita, M. (2008). Is Mongolia's groundwater increasing or decreasing? The case of the Kherlen River basin. Hydrological sciences journal, 53(6), 1221-1229. doi:10.1623/hysj.53.6.1221

[Figure]

Amur-Argun-Kherlen River System. The Amur river network is drawn in blue, the Kherlen River in black, while their boundaries are in green and grey, respectively. The orange line represents the basin boundary of the Amur River basin from GRDC.

Besides, to illustrate the differences between GRDC and the 12D network over arid regions, the example of the Amur River basin has been replaced by the Tigris-Euphrates river system. The text and Fig. 8 with its caption are modified accordingly.

[Figure]

Figure 8. Tigris-Euphrates river system.

*L204: "This major difference can be neglected since it is within the arid region of the Arabian Peninsula."*
*This sentence is confusing, I assume the authors are discussing about the Amur basin, but why "Arabian Peninsula" is mentioned?*

Of course, there was a mistake here. Nevertheless, we now show the example of the Tigris-Euphrates river system, and the portion of the basin which differs between river networks is in the Arabian Peninsula.

*L328: "we prefer here to focus on the routing part and capillary rise as well as floodplain evaporation deactivated."*
*It must be better to note that floodplain scheme affect river discharge and thus evaluation metrics is also affected. Therefore, there are some uncertainties in stating "increase in evaluation metrics meand river network quality is better". This point must be discussed.*

We agree that evaluation metrics are affected by both the river and floodplain processes (as well as groundwater processes). Nevertheless, in this section, we aim at evaluating the whole set of new parameters, including those related to rivers and those related to aquifers and floodplains. To that purpose, we use the CTRIP model which accounts for all these processes (each of them has been validated separately in previous studies). With these considerations, we agree that improved performances are not necessarily due to a better representation of the river dynamics (river network and parameters) but to an overall better representation of rivers, floodplains and aquifers dynamics. That said, we think the pointed sentence could be confusing and has been rephrased as: "Although the ISBA and CTRIP models are fully coupled in Decharme et al. (2019), we prefer here to run the CTRIP model in offline mode; then the configuration considered here includes the representation of floodplains and aquifers, but backward fluxes to ISBA (capillary rise and evaporation over floodplains) are neglected."

*L406: "impacted by the new parametrization"*
*What the authors mean by "new parameterization"? Does this simply mean "new river network map" or does this mean "sub-grid topography parameters"? Please clarify.*

As stated in the previous answer, the evaluation metrics are impacted by the new representation at 12D of river routing (river network and parameters), floodplains (roughness and sub-grid topography) and groundwater (aquifer parameters and sub-grid topography). This has been clarified in the revised version.

*L419: "which impacts the generation of floodplains and aquifers sub-grid parametrization; 2. the use of observed-based river width for CTRIP-12D."*
*I assume floodplain scheme is deactivated for these simulations, so it is not reasonable to discuss its potential impacts on simulation here. Also, did ground water scheme considered in this test simulations? Please provide informations.*

We understand that there was a misunderstanding of the CTRIP configuration used for the simulations at HD and 12D. Floodplain and aquifer schemes are activated here, only the feedbacks to the ISBA model (capillary rise and evaporation over floodplains) are neglected. As stated previously, we tried to clarify this in the description of the modelling configuration (section 4.1).

We thank again the reviewer for his comments which, we think, helped us to improve the manuscript.

---

## Author Comment (AC3)

essd-2021-434

**River network and hydro-geomorphology parametrization for global river routing modelling at 1/12° resolution**

Simon Munier and Bertrand Decharme

Author response to reviewer #3

*Reviewer comments are in italic and blue font.*

**General comment**

*In this work, Munier and Decharme reported a new global global-scale river network 1/12°, which was derived from the widely used MERIT-Hydro dataset. High spatial resolution river networks are increasingly important for current/future studies on water resources management, climate impact on hydrological processes (e.g., floods), etc. The updated river network represents a great advance in delineating global stream networks, although it is derived from previous datasets and models. In addition, the authors also derived a set of hydro-geomorphological parameters, which would facilitate future studies on network-based hydrological and geomorphological, or even biogeochemical studies (like greenhouse gas emissions from streams and rivers). This updated river network map with higher spatial resolution is thus quite important and will greatly contribute to the scientific community. The manuscript was well organized, but some details were missing/lost, which should be addressed (see specific comments below).*

We would like to thank the anonymous reviewer for his/her valuable comments on the manuscript. Bellow are the responses to all the comments raised by the reviewer.

*My another concern is related to validation and data quality. The authors have tried to compare the new river network with previous network datasets or models. To help readers to follow, it may be clearer to present the comparison results (e.g., % in differences) in a table so that readers can easily find out the improvements or performance of this new river network product. This also applies to the derived hydro-geomorphological parameters, in particular for the groundwater and floodplain components, which warrant further data quality assessments.*

As specified in Section 2.3 "Quality assessment", the newly derived river network has been assessed qualitatively and quantitatively over the 69 largest basins of the world. The quantitative assessment is done by comparing the basin area from different sources and the relative difference with the new river network, and by computing the IoU index (Eq. (1), L183) of the basin masks. As written in L186, details of the statistics are gathered in Table S1 in supplementary material. Moreover, in this table, possible causes of main differences are identified.

Concerning the derived hydro-geomorphological parameters describing the groundwater and floodplain components, a direct quantitative assessment is not possible since there is, to our knowledge, no equivalent existing dataset at the same spatial resolution. This is why we proposed an indirect assessment using the CTRIP model, which has been extensively validated in previous work (see Decharme et al., 2019, and references therein), especially in its groundwater and floodplain components. This has been clarified in the revised version of the manuscript.

*Specific comments (with line number):*

*L117: what's the difference between pixel and cell?*

As written in L117-118, a pixel is a unit element at high resolution (1/1200°) while a cell is a unit element at the 12D resolution (1/12°).

*L137: if I understand correctly, 1000 pixels, if near the equator, is ~8.1 km2. It might be reasonable to assume a headwater stream develops within this area size in temperate regions. But for tropical regions, I'm afraid this threshold is too large (i.e., more than 1 headwater stream has developed in 8.1 km2) while for arid regions, the threshold is too small (i.e., a headwater stream may have not necessarily developed within 8.1 km2).*

Yes, the reviewer is right, a threshold of 1000 pixels corresponds to different areas depending on the latitude. Also, the real size of headwater streams may depend on the region.
Yet, the type of river network required by most of river routing models (especially those working with the D8 convention) has to provide a flow direction for each cell of the model. This ensures the closure of the global scale water budget. The type of soil (nature, river, lake, cities etc.) and other characteristics (such as climate zone) are then not considered to set up the global scale river network.
Consequently, the threshold of 1000 pixels is only used to ensure that the considered river drains at least 10 % of the cell. In that sense, the river network should be considered as a drainage network. This has been clarified in the revised version.

*L150: what's a D∞, please explain.*

As stated in L52, Dinf is a drainage network convention for which the water in a unit catchment may flow into any other unit catchement (not necessarily a neighouring one). In this sentence, Dinf has been changed to D∞.

*Fig 3: the figure caption is repetitious. It is not necessary to repeat the text already shown in the text.*

The figure caption has been changed to:
Figure 3. Example of river diversions within the Loire River basin (France). As in Fig. 2, rivers are treated in descending order of their drainage area: 1. the Loire river (dark blue), 2. the Vienne river (light blue), 3. the Cher river (green), 4. the Creuse river (orange) and 5. the Indre river (red). Solid lines and dashed lines represent rivers at HR and 12D, respectivelly. Green squares represent gauge stations.

*L160: have you assessed the error resulting from such diversion processing?*

No, the error resulting from the diversion processing has not been assessed rigorously. The main error caused by the diversion processing relies in the attribution of runoff generated by a Land Surface Model (LSM) to wrong cells of the river network. Yet, the current spatial resolution of most LSMs is generally greater than 1/12° (usually 0.25° or larger at global scale), which suggests that runoff fields would not show high spatial variability at 12D, then minimizing the diversion error.

On the other hand, without the diversion processing, some rivers may merge at wrong locations, causing potential large errors in the river network structure, as show in the following figure. This figure will be integrated as a part of Fig. 3.

[Figure]

Figure: Schematic representation of the structure of the part of the Loire river network shown in Fig. 3.

*Fig 5: This global river network map is nice. But the delineated network results in some regions may be problematic, including Greenland, the Sahara desert, and perhaps the middle east (Saudi Arabia). The high river density in these regions is inconsistent with the real world. Also, in fig 6, why are there rivers in the Great Lakes in the USA/Canada?*

As stated in a previous answer, here the river network should be seen as a drainage network, and in that sense, it has to provide a flow direction for every continental cell, no matter the type of soil or any climatological characteristic. This is required to ensure the closure of the global scale water budget in Earth System Models.
The mean runoff used in section 2.3 to determine arid regions could be used to mask out the river network in such regions, but we preferred here to show the entire drainage network. Also, for the same reason, lakes are not considered in the drainage network. Instead, lakes can be integrated into the river network for models able to simulate the water budget within lakes (see e.g., Guinaldo et al., 2021).

*L174: change 'consists in' to 'consists of', also in L353*

Done, thanks.

*L228: could river channel slope be estimated for each cell? With only one elevation for a cell, how could the slope be calculated? Please clarify.*

For a given cell, we consider the corresponding HR river stretch to compute the river slope as the difference between the elevations of the first and last pixels of the HR river stretch divided by the its length. This has been added in the revised version.

*L235: change 'contrarily' to 'contrary'*

Done.

*L275: refs??  Also, add 'a' before 'number of….'.*

"refs" has been removed, and 'a' has been added.

*L320-322: references are missing.*

The following references has been added: Alkama et al. (2010), Decharme et al. (2012, 2019), Vergnes et al. (2012, 2014).

*L329: 'nearest'*

Done.

---

## Author Comment (AC4)

essd-2021-434

**River network and hydro-geomorphological parameters at 1/12° resolution for global hydrological and climate studies**

Simon Munier and Bertrand Decharme

Author response to reviewer #4

*Reviewer comments are in italic and blue font.*

*This paper uses the DRT method to upscale MERIT-Hydro hydrography datasets to 1/12 degree, and used it for CTRIP streamflow simulations and compare it with a coarse resolution CTRIP run. While this work is very interesting and involves lots of work, I found it lacking sufficient justification to be published in ESSD, as this journal focused more on "data" instead of "model simulation".*

We thank the reviewer for his/her valuable comments. Bellow are our answers to each comment.

*"First, the 1/12 degree river network data and its hydro-geomorphology data seems to be specifically designed for CTRIP and I am wondering what is the wider use of this dataset for other models."*

The main purpose of this paper is to present the global river network at 1/12° and corresponding consistent hydro-geomorphological parameters. This dataset is mainly designed for all global or regional scale grid-based river routing models (RRMs), although it could be used in a variety of hydrology-related studies that need flow direction at a medium spatial resolution (see, e.g., Catalán et al., 2016; Robinne et al., 2018; Scherer et al., 2018; Wan et al., 2015; Zhou et al., 2015). A majority of large-scale RRMs uses a gridded structure for global hydrological studies (see technical review of Kauffeldt et al. 2016) and most of them are still running at a coarse spatial resolution. So with the entire dataset described here (flow direction, river length, river slope, river bank-full depth, river roughness, floodplains roughness, major groundwater basins boundaries, aquifer transmissivity, and aquifer effective porosity), many hydrological models could improve their river routing module by increasing the spatial resolution. Moreover, this consistent and comprehensive dataset can help the modellers to integrate some important processes (such as inundation and groundwater) that are still neglected in some models.

For clarity, we change the title to: "River network and hydro-geomorphological parameters at 1/12° resolution for global hydrological and climate studies".
Also, we added the following references in L26: Arora and Boer (1999), Getirana et al. (2021), Guimberteau et al. (2012), Schrapffer et al. (2020).

Arora, V. K., & Boer, G. J. (1999). A variable velocity flow routing algorithm for GCMs. Journal of Geophysical Research: Atmospheres, 104(D24), 30965-30979.

Catalán, N., Marcé, R., Kothawala, D. N., & Tranvik, L. (2016). Organic carbon decomposition rates controlled by water retention time across inland waters. Nature Geoscience, 9(7), 501-504.

Getirana, A., Kumar, S. V., Konapala, G., & Ndehedehe, C. E. (2021). Impacts of fully coupling land surface and flood models on the simulation of large wetlands' water dynamics: The case of the Inner Niger Delta. *Journal of Advances in Modeling Earth Systems*, 13, e2021MS002463. https://doi.org/10.1029/2021MS002463

Guimberteau, M., Drapeau, G., Ronchail, J., Sultan, B., Polcher, J., Martinez, J.-M., Prigent, C., Guyot, J.-L., Cochonneau, G., Espinoza, J. C., Filizola, N., Fraizy, P., Lavado, W., De Oliveira, E., Pombosa, R., Noriega, L., & Vauchel, P. (2012). Discharge simulation in the sub-basins of the Amazon using ORCHIDEE forced by new datasets, Hydrol. Earth Syst. Sci., 16, 911–935, https://doi.org/10.5194/hess-16-911-2012

Kauffeldt, A., Wetterhall, F., Pappenberger, F., Salamon, P., & Thielen, J. (2016). Technical review of large-scale hydrological models for implementation in operational flood forecasting schemes on continental level. Environmental Modelling & Software, 75, 68-76.

Robinne, F. N., Bladon, K. D., Miller, C., Parisien, M. A., Mathieu, J., & Flannigan, M. D. (2018). A spatial evaluation of global wildfire-water risks to human and natural systems. Science of the Total Environment, 610, 1193-1206.

Scherer, L. A., Verburg, P. H., & Schulp, C. J. (2018). Opportunities for sustainable intensification in European agriculture. Global Environmental Change, 48, 43-55.

Schrapffer, A., Sörensson, A., Polcher, J., & Fita, L. (2020). Benefits of representing floodplains in a Land Surface Model: Pantanal simulated with ORCHIDEE CMIP6 version. Climate Dynamics, 55(5), 1303-1323.

Wan, Z., Zhang, K., Xue, X., Hong, Z., Hong, Y., & Gourley, J. J. (2015). Water balance-based actual evapotranspiration reconstruction from ground and satellite observations over the conterminous U nited S tates. Water Resources Research, 51(8), 6485-6499.

Zhou, Y., Hejazi, M., Smith, S., Edmonds, J., Li, H., Clarke, L., ... & Thomson, A. (2015). A comprehensive view of global potential for hydro-generated electricity. Energy & Environmental Science, 8(9), 2622-2633.

*"Second, a larger portion of this study is on comparing two simulations of CTRIP runs, instead of focusing on the river network dataset."*

In the present form of the manuscript, 13 pages over 23 pages for the main text focuses on the river network and hydro-geomorphological dataset, and 13 Figures over 18 Figures. The section 4 (CTRIP runs) uses 7 pages and the remaining figures. So we do not consider that a larger portion of this study is on comparing two simulations of CTRIP runs. This section 4, where the CTRIP simulations are presented, should be seen as a validation of the 12D dataset. The detailed CTRIP modelling configuration and validation can be found in other articles (e.g., Decharme et al., 2019, and references therein). Given the known quality of the CTRIP model at 0.5° resolution, we think that the overall improvement from this coarse resolution to 1/12° resolution is a good indicator of the overall quality of the dataset presented in this manuscript. Validating the different parameters derived in this study is not possible at the global scale because of lack of observed data. Hence, we chose to validate the entire dataset in the context of river routing modelling with the CTRIP model as an example. We argue that most RRMs use (or go to use) similar parametrization (river network, river length, width and slope, roughness, etc.) and could benefit from this dataset, built to ensure the consistency between the parameters.

*"The authors seem to not have introduced new updates to DRT. So I cannot help asking what is their "data contribution"? It seems an existing method (DRT) was applied to an existing dataset (MERIT Hydro). To justify its publication in ESSD, I think authors will need to make more efforts to describe their contribution to data (instead of to model simulation)."*

As noted by the reviewer, our 12D dataset is built by applying an existing method (DRT) to an existing dataset (MERIT Hydro). Note that our upscaling algorithm is slightly different than the one from Wu et al. (2012), for instance in the river diversion processing, in the treatment of estuaries or in the fact that rivers are treated hierarchically instead of basins. Besides, the dataset we provide does include not only the newly developed river network but also the associated fully consistent set of hydro-geomorphological parameters. We then consider that this dataset is a new product that, we think, could be useful for other RRMs and as such, deserves to be published.

*So I cannot recommend publish this paper unless these questions are sufficiently addressed.*

We hope that our previous and following answers will convince the reviewer of the usefulness of our dataset.

*Here are more comments:*

*FIG. 15: Can the authors mention it is daily or monthly evaluation? Can you add both daily and monthly evaluation here? Because routing models generally matter more for daily streamflow simulations than monthly. If it is monthly then L385 "clearly shows quite good performances" should be revised a little bit.*

We agree that RRMs are generally more focused on daily streamflows. In this validation section, simulated discharges are compared to observed discharge only at a daily time step. This has been clarified in the revised version.

*Fig. 16: I do not think this figure is separately needed. Because there is not much information in the main text (around L384), it can be added as a subplot to Fig. 15. Otherwise, authors should describe much more about Fig. 16 to justify the use of this figure.*

We agree. Fig. 16 has been added as a subplot to Fig. 15.

*L405 and Fig. 17: why only show stations with KGE > -1? Didn't CTRIP-12D do better than HD for all KGEs? This is a bit confusing and needs more description.*

Despite the overall good quality of the CTRIP model, it may fail in reproducing observed discharges (arbitrarily KGE < -1), in particular for stations highly influenced by human activities which are not represented in CTRIP. For these stations, we consider that the CTRIP model is not adapted due to processes not accounted for. Consequently, we consider that improvement or degradation of model performances are not relevant and we discarded these stations. Note that considering these stations leads to the same result (70 % improvement, 30 % degradation). This has been clarified in the revised version.

*Also, about the use of NIC, why not simply use KGE differences, and positive KGE means better performances?*

The advantage of the NIC criterion is that it normalizes the difference between the KGE of two experiments. A given KGE difference has not the same impact in terms of performance depending on the value of KGE. For instance, if KGE_ref=0 and KGE_new=0.2 then NIC=0.2, whereas if KGE_ref=0.8 and KGE_new=1 then NIC=1. The higher NIC value in the second case means that the improvement is better (perfect in that case) although the difference is the same. This has been clarified in the text.

*Fig. 18: again, this figure has little new information than Fig. 17. L408 says "a closer look at" but I didn't think this adds much information other than saying the same thing as Fig. 17.*

The last paragraph of section 4.3.2 has been modified as:

Better performances could be expected for smaller basins since these basins are represented by just a few cells at HD, and the difference between the basin delineation at HD and 12D could be relatively high, then leading to different contributing areas. The better performances of CTRIP-12D for larger basins is less expected. Indeed processes and forcing are the same for both configurations and parameters are derived using similar strategies and relationships. The improvement of the correlation and variability demonstrates that a better defined river network improves the dynamics of river propagation within the basin and interactions with floodplains and aquifers.
Other potential sources of differences between both models include: 1. the reference HR dataset (HydroSHEDS for CTRIP-HD, MERIT-Hydro for CTRIP-12D), which impacts the generation of floodplains and aquifers sub-grid parametrization; 2. the use of observed-based river width for CTRIP-12D.

This has been corrected. Thanks.

---

## Author Response (AR2)

**River network and hydro-geomorphological parameters at 1/12° resolution for global hydrological and climate studies**

Simon Munier and Bertrand Decharme

Reviewer comments are in italic and blue font.

**Author response to reviewer #3**

*The authors have substantially improved the manuscript. But for the validation and data quality (one of my major concerns), in the revised version, the authors did not critically evaluate the improvement or differences from previous products (e.g., GRDC, DRT as shown in Table S1). The authors stated that 'only a few basins show a relative difference greater than 10 %' (L216). This is actually not the case. For example, for the GRDC, the difference is 13%, and it is even 23% for the DRT. For some rivers, the difference could be as high as 95%.... I believe the authors should be able to address these large differences in the revised version.*

A lot of information are gathered in Table S1 on the differences between basin delineation from the different river networks. We highlighted the basins showing the largest differences and tried to identify the causes of all these differences. We then rewrote some parts of section 2.3 (Quality assessment) and added a more comprehensive analysis of Table S1 in the text (see bellow). We also added some statistics (median and mean) in Table S1. We think that now, all the major differences are addressed, and we thank again the reviewer for his/her valuable comment.

New text in section 2.3 Quality assessment:

[revised manuscript text omitted]

**Author response to reviewer #4**

*The authors have mostly satisfactorily addressed my comments. One minor comment remained, which should not influence the decision: in the Conclusions section, it should be helpful for authors briefly mention the difference between vector-based and grid-based RRMs (as authors have already done in the Intro), and that these 1/12-degree river network datasets are the most helpful to high-resolution grid-based models (and make a few examples on which grid-based models can benefit from this the most). This way readers can fully appreciate the wider applicability of their data contribution, and also clear the confusion on why not directly supplying readers with vectors.*

We would like to thank the reviewer for this remaining comment. We rewrote the last sentence of the conclusion as follows.

In grid-based approaches, the river network is discretized on a regular Cartesian grid, so that unit-catchments are rectangular pixels with their own hydrogeomorphological characteristics. The complete dataset described here is particularly well suited to a number of large-scale RRMs using a gridded structure for global hydrological studies (see Table 2 in Kauffeldt et al. , 2016). Not all of them are currently running at 12D resolution, while, on the other hand, the current tendency suggests that 5 arcmin could become the next standard resolution for global scale climate studies, namely via the release of the last global meteorological dataset for impact models in phase 3a of the Inter-Sectoral Impact Model Intercomparison Project (ISIMIP3a, Dirk et al. , 2022). With the entire dataset described here (flow direction, river length, river slope, river bank-full depth, river roughness, floodplains roughness, major groundwater basins boundaries, aquifer transmissivity, and aquifer effective porosity), many hydrological models could improve their river routing module by increasing the spatial resolution. Moreover, this consistent and comprehensive dataset can help modellers to integrate some important processes (such as inundation and groundwater) that are still neglected in some models.

Dirk N. Karger, Stefan Lange, Chantal Hari, Christopher P. O. Reyer, Niklaus E. Zimmermann (2022): CHELSA-W5E5 v1.0: W5E5 v1.0 downscaled with CHELSA v2.0. ISIMIP Repository. https://doi.org/10.48364/ISIMIP.836809.3

---

## Author Response (AR3)

essd-2021-434

**River network and hydro-geomorphological parameters at 1/12° resolution for global hydrological and climate studies**

Simon Munier and Bertrand Decharme

In this last version of the manuscript, all the references have been revised to meet the requirements of the journal.

In addition, a new version of the dataset has been released in zenodo, which now includes the source code for the derivation of the upscaled river network and the hydro-geomorphological parameters.